# Mechanical Properties of Short Fiber-Reinforced Geopolymers Made by Casted and 3D Printing Methods: A Comparative Study

**DOI:** 10.3390/ma13030579

**Published:** 2020-01-26

**Authors:** Kinga Korniejenko, Michał Łach, Shih-Yu Chou, Wei-Ting Lin, An Cheng, Maria Hebdowska-Krupa, Szymon Gądek, Janusz Mikuła

**Affiliations:** 1Institute of Materials Engineering, Faculty of Material Engineering and Physics, Cracow University of Technology, Jana Pawła II 37, 31-864 Cracow, Poland; michal.lach@pk.edu.pl (M.Ł.); maria.hebdowska-krupa@mech.pk.edu.pl (M.H.-K.); szymon.gadek@mech.pk.edu.pl (S.G.); jamikula@pk.edu.pl (J.M.); 2Department of Civil Engineering, National Ilan University, No.1, Sec. 1, Shennong Rd., I-Lan 260, Taiwan; siyuzou@gmail.com (S.-Y.C.); wtlin@niu.edu.tw (W.-T.L.); anfendib02@gmail.com (A.C.)

**Keywords:** fly ash-based geopolymer, 3D printing, additive manufacturing, short fiber, flax fiber, carbon fiber

## Abstract

The main objective of this article is to develop ceramic-based materials for additive layer manufacturing (3D printing technology) that are suitable for civil engineering applications. This article is focused on fly ash-based fiber-reinforced geopolymer composites. It is based on experimental research, especially research comparing mechanical properties, such as compressive and flexural strength for designed compositions. The comparison includes various composites (short fiber-reinforced geopolymers and plain samples), different times of curing (investigation after 7 and 28 days), and two technologies of manufacturing (casted and injected samples—simulations of the 3D printing process). The geopolymer matrix is based on class F fly ash. The reinforcements were green tow flax and carbon fibers. The achieved results show that the mechanical properties of the new composites made by injection methods (simulations of 3D technology) are comparable with those of the traditional casting process. This article also discusses the influence of fiber on the mechanical properties of the composites. It shows that the addition of short fibers could have a similar influence on both of the technologies.

## 1. Introduction

In the 21st century, cementless blended composites can be used in construction materials to reduce CO_2_ emissions and preserve natural resources [1]. This is a significant and innovative topic for material research. In accordance with resource sustainability, these composites should be used in eco-friendly or environmentally friendly materials to replace the cement. There is an increasing interest in the use of cementless blenders (known as alkali-activated binders) and geopolymers. Geopolymers are amorphous three-dimensional alumina–silicate binder materials that can be synthesized by mixing alumina–silicate reactive materials with strong alkaline solutions [2]. Several industrial wastes or by-products have been used for the production of geopolymer composites, such as fly ash, slag or metakaolin [3,4]. The geopolymers provide cementless blenders that are more competent and perform better than the conventional cement-based composites [5]. In fact, the advantages of geopolymers can be capitalized on by reusing by-products or waste as full cement replacement for binding materials. This development in construction materials can improve the greenness and sustainability of normal cement-based composites and help to maintain comparable and even superior properties [6]. According to statistics, the manufacture of geopolymeric binders is expected to produce 80% less CO_2_ emissions than that of ordinary Portland cement. The primary reason for this is that the limestone does not need to be calcined to produce the geopolymeric binder [7,8].

In general, the inclusion of fiber in cement-based composites can reduce the growth or propagation of internal cracks, and help to transfer load known as fiber-reinforced cement-based composites [9]. The fiber-reinforced composites have higher tensile strength, flexural strength and ductility than ordinary cement-based materials. They demonstrate a significant increase in energy absorption and toughness, and the fibers in cement-based composites help to reduce permeability in cracked concrete specimens. Considerable developments have been made on geopolymeric mortar or concrete, which has been made to replace the cement-based binder in the current fiber-reinforced cement-based composites. It acts as a geopolymeric binder, known as fiber-reinforced geopolymers, and results in higher tensile strength, tensile capacity, and durability [10,11,12]. The research field of fiber-reinforced geopolymers is an important application and the field is still relatively young.

Nowadays, additive manufacturing (AM) is a rapidly developing industrial sector and a disruptive technology. It is also an answer to new challenges such as resource saving and energy effectiveness, as well as a response to circular economy needs [13,14]. The technology of 3D printing brings low labor costs, less waste, and high efficiency. Unfortunately, its application is limited, and only some prototype elements have been performed using this technology. The full exploitation of 3D printing technology for ceramic materials is currently a great challenge due to the in-process and in-service performance of the available materials’ sets, especially their application in the construction industry [15,16].

Further development of this technology requires improvements to design new materials such as geopolymeric binders and fiber-reinforced geopolymers [17,18]. Research into 3D printing technology in the case of concretes has been done, but investigation into other ceramic materials such as geopolymers is still new in the field [19,20,21]. Some researchers have decided to include short fibers in cementitious materials to achieve desired toughness and ductility enhancement for 3D printing [22,23,24]. This results in sufficient mechanical capacities such as rebar reinforced structures.

So far, only a few studies have been provided in the area of reinforced 3D-printed composites using short fibers including short steel fibers [25], short glass fibers [23], and different kinds of plastic fibers [25,26].

Short steel fibers were investigated by Al-Qutaifi et al. [25]. The samples were based on fly ash with 1% steel fibers (40 mm length) and 0.5% polypropylene (PP) fibers (5 mm length), prepared by simulating the extrusion method process. The results of the 3-point bending test show that the flexural strength increased for reinforced samples. It was 4.99 MPa for samples without reinforcement, 6.31 MPa with steel fibers, and 5.13 MPa with glass fibers [25]. The research shows that using steel fibers as a reinforced material in 3D printing could also have negative consequences for impedance and the complete adhesion between additive layers [25]. It suggests that other materials should be tested as reinforcement in 3D printing technology.

The incorporation of short glass fibers in geopolymer matrixes was studied by Panda et al. [23]. They investigated three different lengths of glass fibers: 3 mm, 6 mm and 8 mm, and additives in different fractions: 0.25%, 0.50%, 0.75% and 1.0%. The main aim of the research was to discover the mechanical properties of the 3D-printed composites [23]. The matrix composition was based on fly ash, slag, silica and fine river sand. The flexural and tensile strengths of composites manufactured by the 3D printing method significantly increased with the addition of fibers. The best results were achieved for 1.0%. In this article, there is a lack of comparison with the samples without fibers [23].

The other investigations were provided by Nematollahi et al. [26]. His research team works on fly ash-based geopolymers with three kinds of polymer fibers: polyvinyl alcohol (PVA), polypropylene (PP), and polyphenylene benzobisoxazole (PBO), each 6 mm in length. The applied addition was 0.25% by volume. The main objective of fiber addition was to reduce the inter-layer bond strength of the 3D-printed geopolymer. The samples were prepared by extruding material and using the traditional casting method. Next, both types of samples were cured at 60 °C for 24 h [26]. The 3D-printed fiber-reinforced geopolymer had better mechanical properties than samples without reinforcement. The flexural strength was 9.0–10.3 MPa and inter-layer bond strength was 2.33–2.58 MPa, depending on the type of fibers. For the material of the matrix, these values were 7.7 MPa and 3.03 MPa, respectively [26].

This study is beyond the state of the art, due to its incorporation of new types of fibers into the geopolymer matrix, using more additive manufacturing technology and complex investigation in this area in comparison with the previous one [27]. The aim of this article is to compare mechanical properties, such as compressive and flexural strength, for designed compositions and short fiber-reinforced geopolymers, using 3D printing injection technology. Two kinds of short fibers were used in the geopolymeric mixture, and this tested the feasibility and applicability of short fiber-reinforced geopolymers made by 3D printing technology.

## 2. Materials and Methods

### 2.1. Materials

The geopolymer matrix was made from fly ash, and a specimen with a water– binder ratio (*w/b*) of 0.35 and a binder–sand ratio of 0.50 was prepared. The fly ash came from a combustion process at the CHP plant in Skawina (Poland). This fly ash has a chemical composition typical for class F [28]. It contains less than 5% unburned material, up to 10% iron compounds, and a very low amount of calcium compounds. It includes about 36% reactive silica and a large amount of amorphous phase [28,29]. The physical parameters are appropriate for geopolymerization. The specific density of fly ash is 2.80 and the content of particles under 45 μm in size is ca. 88% [28,29]. It contains a large amount of spherical particles, and because of that its workability is good [29]. It was adjusted with a constant dosage of NaOH and sodium silicate (water glass) for use as an alkali activator, respectively. The activator consists of an aqueous solution of sodium hydroxide with 10 M, and an aqueous solution of sodium silicate, in a ratio of 1:2.5. The specific densities and fineness moduli of the sand samples are 2.65 and 2.84, respectively.

The green tow flax fibers were purchased from the Institute of Natural Fibers and Medicinal Plants in Poland, Poznań. Tow is a coarse, broken fiber, removed during flax processing. These fibers are usually shorter than 30 cm and are semi-products in textile fiber production [30]. The fibers used as samples in this study were 30–50 mm in length (Figure 1a). The properties of the carbon fibers, declared by the producer, used in this study were density between 1.6 and 2.0 g/m^3^, Young’s modulus of 230 GPa, tensile strength between 2800 and 5000 MPa, and elongation in the range 1–1.5%. Carbon fiber has a length of 5 mm and a diameter of 8 μm. The appearance of carbon fiber is shown in Figure 1b.

Three kind of mixtures were prepared: geopolymers without fibers, geopolymers with 1% (by mass) green tow flax fibers, and geopolymers with 1% (by mass) carbon fibers.

### 2.2. Specimens

All specimens were prepared using sodium promoter, fly ash and sand. Half of them were prepared using 1% (by mass of the total composites) flax fibers and carbon fiber. The process of activation used sodium hydroxide solution combined with sodium silicate solution. To produce geopolymers, an aqueous solution of sodium silicate (R-145) with a molar module of 2.50 and a density of about 1.45 g/cm^3^, flakes of technical sodium hydroxide, and tap water were all used. The alkaline solution was prepared by pouring the aqueous solution of sodium silicate over the solid sodium hydroxide. The solution was mixed and left until its temperature became stable and the concentrations equalized, which took about 2 h. The fly ash, sand, alkaline solution and fibers were mixed for about 10 min by using a low speed mixing machine to create the homogeneous paste.

Next, three methods of production were implemented: Type I—traditional pouring molding using a vibrating table; Type II—injection molding made to simulate 3D printing using a vibrating table; and Type III—traditional pouring molding without using a vibrating table. The appearances of the compressive and flexural strength made by the injection method are illustrated in Figure 2.

The geopolymer concrete specimens were cast into a 50 mm × 50 mm × 50 mm cubic mold for the compressive strength. The specimens for flexural strength were casted into a 50 mm × 50 mm × 200 mm prismatic mold. After casting, the specimens, together with the molds, were cured in a laboratory dryer at 75 °C for 24 h. After demolding, the geopolymer concretes for standard curing were placed in the standard curing room for 7 and 28 days.

### 2.3. Testing Methods

#### 2.3.1. Density

The dimensions were calculated according to the length of the laboratory caliper, and the weight was determined by the laboratory analytic weight. The composites do not have significant porosity, and because of this, the calculation was made for solid, non-porous materials.

#### 2.3.2. Compressive Strength

The compressive strength of cubic specimens was tested in accordance with the methodology described in the standard EN 12390-3, after 7 and 28 days. The average results of six specimens of each group were reported, and the testing process was performed on a concrete press, the MATEST 3000 kN with a speed of 0.5 MPa/s. The tests involved cubic samples with the dimensions 50 mm × mm 50 × 50 mm. The tests involved at least 6 samples (in total, 6 types of production methods × 6 compositions × 2 periods = 72 samples).

#### 2.3.3. Flexural Strength

Flexural strength tests were carried out according to the methodology described in the EN 12390-5 standard. The average results of three specimens of each group were reported, and the testing process was performed using a universal testing machine, the MATEST 3000 kN (MATEST S.p.A., Arcore, Italy) with a speed of 0.05 MPa/s. The tests involved prismatic samples with the dimensions 50 × 50 × 200 mm (the space between the supporting points was 150 mm). The tests involved at least 3 samples (in total, 3 types of production methods × 3 compositions × 2 periods = 18 samples).

## 3. Results

### 3.1. Density

The results for the material without reinforcement were 1.60 g/cm^3^ after 7 days, and 1.62 g/cm^3^ after 28 days. The samples reinforced by flax fiber achieved values 1.54 g/cm^3^ and 1.48 g/cm^3^ respectively. The density of the samples reinforced by carbon fibers was 1.67 g/cm^3^ after 7 days, and 1.58 g/cm^3^ after 28 days. The differences in density were not significant for different types of composition. There is no direct correlation between the density and the results of mechanical properties for this series of samples.

### 3.2. Compressive Strength

The results of the compressive strength tests for each mixture are illustrated in Figure 3, Figure 4 and Figure 5. For the specimens without fibers as shown in Figure 3, the compressive strength of specimens made by the injection method (simulation of the 3D printing process) was similar to the strength of those made by the traditional casting method. Proper flowability was useful for molding in the traditional casting or injection method.

The compressive strength of specimens containing flax fibers is shown in Figure 4. It indicates that the specimens made by the injection method had greater strength than those made by the traditional method.

The specimens containing fibers had 6% higher compressive strength than those without fibers. It is noted that the compressive strength was little influenced by the presence of the short flax fibers embedded in the geopolymer, due to the resisting crack under axial loading.

The compressive strength of specimens containing carbon fibers is shown in Figure 5. At the age of 28 days, there was a significant drop in strength of the specimens made by the Type II and III method. The drop in strength was up to 10% more than the specimens made by the Type I method. This may be due to fiber balling or air bubbles in the molding process. Overall, the geopolymers or the short fiber-reinforced geopolymer specimens made by the injection method had close or higher compressive strength than those made by the traditional casting method. On the basis of the previous study, an increase in compressive strength was observed in short fiber-reinforced geopolymers containing 0.5% PP fibers. Beyond this fiber content, the rate of increase of compressive strength decreased at each age [30]. It is evident that a suitable amount of fiber in geopolymers can help to enhance their compressive strength. The optimal dosage of fiber might be 0.5% by mass of the total composites.

### 3.3. Flexural Strength

The results of the flexural strength tests for each mixture are illustrated in Figure 6, Figure 7 and Figure 8. For all mixtures, the specimens containing both fibers had higher flexural strength than those without fibers.

At the age of 28 days, the specimens containing flax fibers made by the injection method had the highest flexural strength—up to 36% more than those made by the traditional casting method (Figure 7).

However, there was no significant increase in the flexural strength of specimens containing carbon fibers, and this could be due to the excessive dosage of carbon fibers (Figure 8). It exhibited about a 36% increase in flexural strength, respectively, along with a significant improvement in post-crack ductility, and the highest toughness of the short fiber-reinforced geopolymers, which is consistent with the previous study [30,31].

The trends of flexural strength are similar to the compressive strength tests, and the short fiber-reinforced geopolymers made by the injection method to simulate for the 3D printing process could be used as a suitable method to develop 3D printing technology.

## 4. Discussion

The 3D-printed composites reinforced by long and short fibers are a relatively new topic. Only a few research studies with long steel fibers [17,18], short steel fibers [25], short glass fibers [24], flax fibers [27], and different kinds of plastic fibers [25,26] have been conducted so far.

As part of their research, Lin et al. [17] created in-process geopolymer reinforcements using a continuous stainless steel cable with three different dimensions: 1 mm, 1.5 mm and 2 mm [17], and used polyvinyl alcohol (PVA) fibers (0.5 wt %, 8 mm) as well. Their research is based on experiments using 3D-printed geopolymer composites reinforced by a steel cable. The results show that a steel cable could improve the flexural strength of 3D-printed material by 290%. Ma et al. [18] investigated micro-reinforcement for a geopolymer–steel cable. The samples were prepared in three different printing paths as incline-crossed, orthogonal-crossed, and rectangular shaped filaments [18]. The investigations confirmed that the most effective sample was the incline-crossed filaments configuration as a reinforcement. The flexural strength of this sample was approximately 48.9%, and 200% higher than alternative materials with reinforcement. It was also 600% higher than samples without reinforcement [18].

Short glass fibers were investigated by Panda et al. [24]. They applied three different lengths of the fibers: 3 mm, 6 mm and 8 mm, and four fractions of the fibers: 0.25%, 0.50%, 0.75% and 1.0% [24]. The flexural and tensile strengths of 3D-printed geopolymers significantly increased with a higher amount of fibers—up to 1.0% [24]. Al-Qutaifi et al. [25] investigated inter-layer bonding, including the influence of time intervals between layers. They researched geopolymers with 1% steel fibers (40 mm length) and 0.5% polypropylene (PP) fibers (5 mm in length). The basic flexural strength was 4.99 MPa with reference samples without reinforcement, 6.31 MPa with steel fibers, and 5.13 MPa with glass fibers [25]. The results also showed reducing time gaps between additive layers [25]. Some preliminary research has also been conducted on flax fibers [27]. The research confirmed the possibilities of using natural fibers as a component of geopolymer composites [27].

Nematollahi et al. [26] investigated three kinds of plastic fibers: polyvinyl alcohol (PVA), polypropylene (PP), and polyphenylene benzobisoxazole (PBO) fibers (each 6 mm in length, and 0.25% in volume). Their research was based on the reduction of the inter-layer bond strength of 3D-printed geopolymers [26]. The results show that the flexural strength of 3D-printed fiber-reinforced geopolymer mixtures was substantially higher than that of the 3D-printed geopolymer with no fiber. The flexural strength was 9.0–10.3 MPa and inter-layer bond strength was 2.33–2.58 MPa depending on the type of fibers. For the reference samples, the average values were 7.7 MPa and 3.03 MPa, respectively [26].

The results indicated that the inclusion of fiber slightly enhanced compressive strength and significantly enhanced flexural strength. It confirmed the research literature. In the case of the research literature, as well as own research, it is shown that the fibers have a positive influence on flexural strength, especially in a long-term perspective. The short fibers, as well as the long ones, are an effective way to reinforce geopolymers. The unexpected finding in this research was the fact that the performance of specimens containing flax fibers was better than that of specimens containing carbon fibers. This requires deeper analysis. However, the inclusion of green tow flax fibers in geopolymers means that they can be shown to be green materials for sustainable development. This topic requires future research, including fiber orientation and distribution change dependent on 3D printing parameters (flow seed, nozzle diameter, fiber length, trajectory etc.), and how it influences the flexural, tensile and compressive strength, as well as fracture energy.

## 5. Conclusions

The compressive and flexural strengths of the specimens made with three different molding methods are comparable to each other for all the analyzed conditions. The results indicate that the inclusion of the fibers slightly enhanced compressive strength, and significantly enhanced flexural strength. The performance of specimens containing flax fibers was better than that of specimens containing carbon fibers. In conclusion, the injection method was an effective simulation for the 3D printing process, and the specimens made by the injection method created better quality products than the traditional casting method. The inclusion of green tow flax fibers in geopolymers means that they can be shown to be green materials for sustainable development. Further research is needed on fiber orientation and the distribution of short fiber-reinforced geopolymers made by the injection method in 3D printing technology.

The development of the AM technologies requires improvements to design new materials, assess materials’ performance and improve processing strategies. The geopolymer material still requires development and optimization. Fiber reinforcement of 3D-printed geopolymers is a promising way to develop the materials for effective application in manufacturing on a larger scale. There could be a promising alternative for concrete in many applications.

## Figures and Tables

**Figure 1 materials-13-00579-f001:**
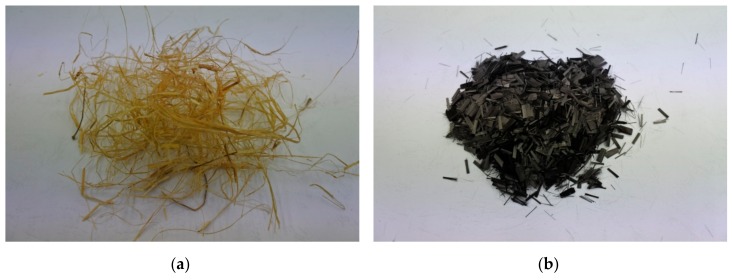
(**a**) The green tow flax fibers; (**b**) The carbon fibers.

**Figure 2 materials-13-00579-f002:**
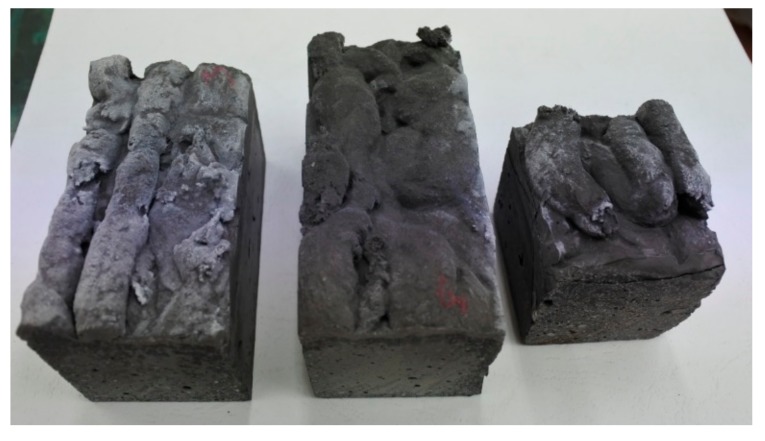
Appearances of the samples made by the injection method.

**Figure 3 materials-13-00579-f003:**
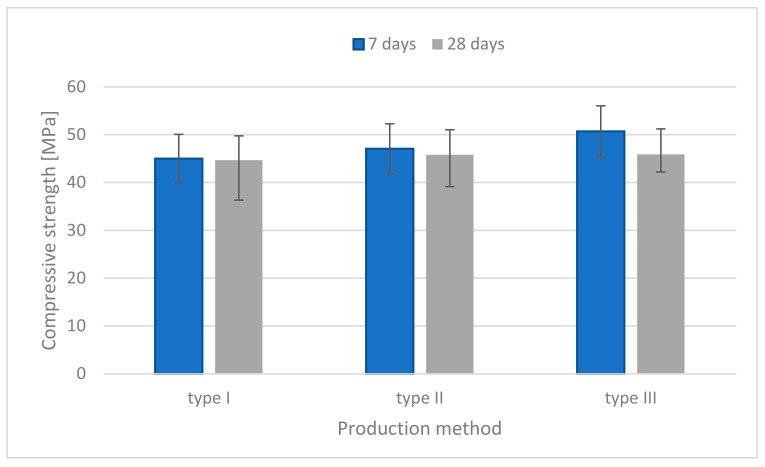
Compressive strength histograms (specimens without fibers).

**Figure 4 materials-13-00579-f004:**
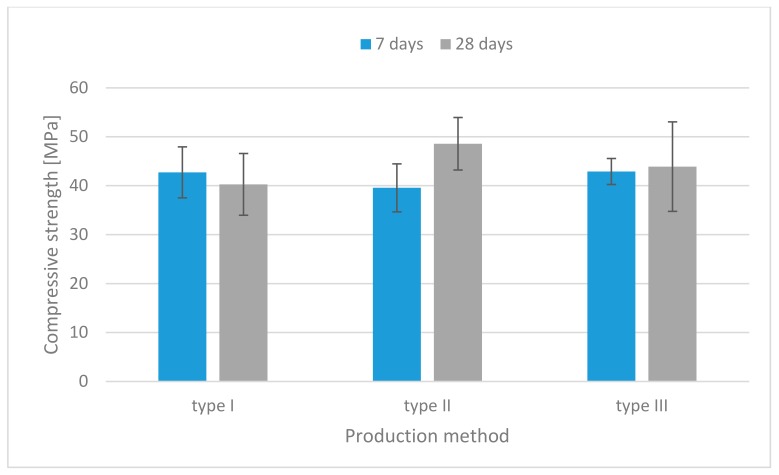
Compressive strength histograms (specimens containing flax fibers).

**Figure 5 materials-13-00579-f005:**
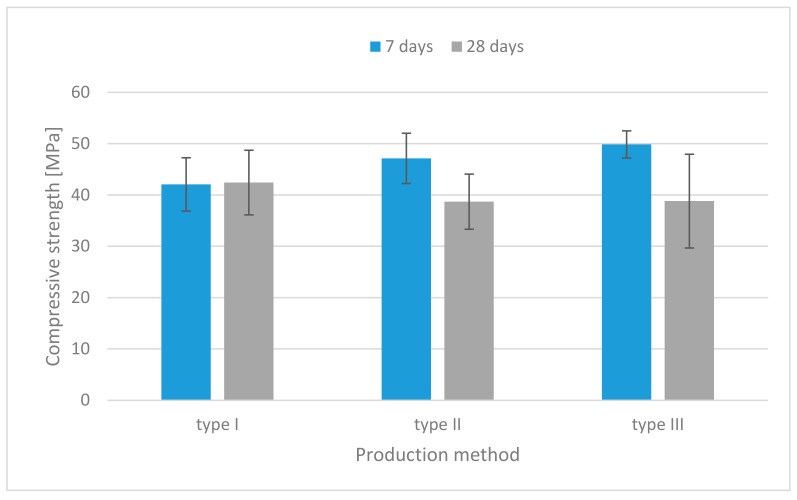
Compressive strength histograms (specimens containing carbon fibers).

**Figure 6 materials-13-00579-f006:**
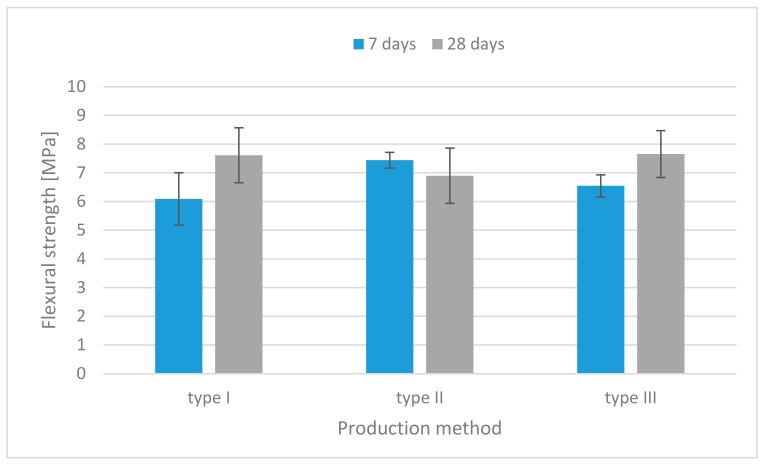
Flexural strength histograms (specimens without fibers).

**Figure 7 materials-13-00579-f007:**
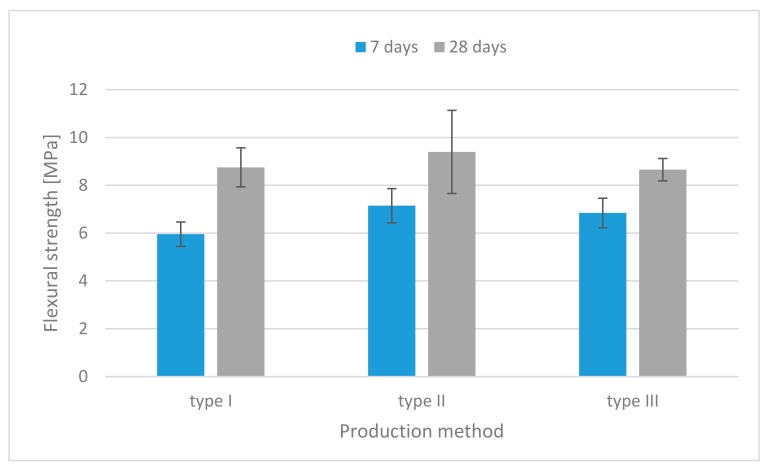
Flexural strength histograms (specimens containing flax fibers).

**Figure 8 materials-13-00579-f008:**
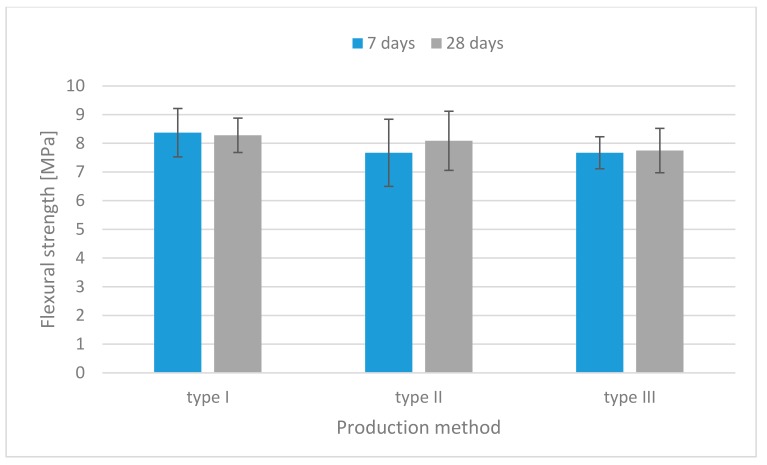
Flexural strength histograms (specimens containing carbon fibers).

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
