# Peer review of "Mechanical Properties of Short Fiber-Reinforced Geopolymers Made by Casted and 3D Printing Methods: A Comparative Study"

_materials, 2020, doi:10.3390/ma13030579_

Round 1

Reviewer 1 Report

The article analysis geopolymers based on short fibre reinforcement and their mechanical performance. The article would be quite interesting if it was written well, but there is a great need to proofread and use English editing services for this article.

Additionally, Introduction section must be improved with what has been already obtained by other scientists (observations, numerical values of mechanical properties, etc.).

Figure 2 has misleading title. How can compressive and flexural strengths be made by injection method? Do you mean specimens for the tests are made by injection method?

All the graphs where the results are presented should contain upper an lower limits with confidence interval of 95% or at least standard deviations.

Please carefully look all the article because 138 line contains the reference to Fig. 6 although it has to be Fig. 3, in my opinion.

Overall, the article is of low quality. It has to be rewritten.

Author Response

The authors thank the reviewer for the kind comments. We appreciate for your recommendation to improve the article for the journal “Materials”. All the comments regarding the reviewer’s comments were carefully addressed, including:

- The introduction has been supplemented.

- The captation under Fig. 2 have been changed. The explanation of the production methods is presented in the text (materials & methods).

- Standard deviation has been added in Figures.

- The references in two places has been corrected (paragraph 3.1. – base version).

Reviewer 2 Report

The paper entitled "Mechanical Properties of Short Fiber-Reinforced Geopolymers Made by Casted and 3D Printing Methods: A Comparative Study" has the potential to be interesting for the scientific community.

Specific comments are as follows:

The Authors probably wrongly call their product - mortar. Their product is a geopolymer (for example lines: 112, 113, 120). Please explain what Fig. 2 shows because the description of the drawing is not consistent with what you see in it. There is no statistical analysis of research results, e.g. ANOVA, logistic regression, etc. Error bars should be provided in Figures 3-8. Only two parameters are shown in this paper: compressive strength and flexural strength. I think that it is too little for the paper to be published in the prestigious journal - Materials.
I am asking for supplementation with other strength or physical tests, such as density, porosity, water absorption, shrinkage, and others. These features are highly correlated with strength and could enrich the paper.

According to my suggestion, the paper can be published, but it needs major restructuration.

Author Response

The authors thank the reviewer for the kind comments. We appreciate for your recommendation to improve the article for the journal “Materials”. All the comments regarding the reviewer’s comments were carefully addressed, including:

- The name of the product has been corrected the product is geopolymer concreto, because the fly ash is mixed with sand. The geopolymer mortar could based only for fly ash The names in the article has been corrected.

- The caption under Fig. 2 have been changed.

- The statistical analysis has been used to calculation the standard deviation.

- The errors bars – standard deviation has been provided in Figures 3-8.

- Density – the article has been supplemented about the results of the density research

- Porosity – the problem of the porsoity has been included in density part.

- The water absorption is not connected with mechanical properties, that are the main point of the intrest in this article. It is utility property of the materials.

- Shrinkage – in case of this materials shrinkage is ca. 0. The samples are stable about dimension. But this property was not a main potin of the article, that is focused on mechanical properties. The shrinckage is material property connected with manufacturing / production proces not with mechanical properties.

Reviewer 3 Report

The manuscript describes the development of the ceramic based materials for additive layer manufacturing (3D printing technology) that are suitable for civil engineering applications.

The introduction must be improved by introduction of a sentence regarding the progress beyond the state of the art due to this research.

The characterization of the geopolymers must be included in the paper. There are necessary proves regarding the formation of the geopolymer.

The results must be improved by introduction of a statistical method in order to clearly demonstrate the existence of missing of differences among different type of samples.

The errors bars must be included in all the figures.

  The compressive and flexural strength seems to be not enough for the comparison of the materials. Please explain.

Author Response

The authors thank the reviewer for the kind comments. We appreciate for your recommendation to improve the article for the journal “Materials”. All the comments regarding the reviewer’s comments were carefully addressed, including:

- The introduction has been supplemented.

- The information about formation of geopolymers is included in section specimens – 2.2.

- The statistical analysis has been used to calculation the standard deviation. Some additional information are provided into research part.

- The errors bars – standard deviation has been added to in Figures.

-The additional research – density has been added to the research part.

-The description of testing methods has been supplemented about the numer of samples.

Round 2

Reviewer 1 Report

Authors have corrected all the remarks.

Reviewer 2 Report

After corrections have been made, the paper is ready for publication.

Reviewer 3 Report

The manuscript was improved and it could be published without further modifications.